# Genetic and physical interactions reveal overlapping and distinct contributions to meiotic double-strand break formation in *C. elegans*

Marilina Raices[1], Fabiola Balmir[1], Nicola Silva[2], Wei Li[1,3], McKenzie K Grundy[4], Dane K Hoffman[1†], Elisabeth Altendorfer[5], Carlos J Camacho[6], Kara A Bernstein[4,7], Monica P Colaiacovo[5], Judith L Yanowitz[1,8]*

[1]Magee-Womens Research Institute, Pittsburgh, United States; [2]Department of Biology, Masaryk University, Brno, Czech Republic; [3]Tsinghua U. Medical School, Beijing, China; [4]Department of Microbiology and Molecular Genetics, University of Pittsburgh School of Medicine, UPMC Hillman Cancer Center, Pittsburgh, United States; [5]Department of Genetics, Blavatnik Institute, Harvard Medical School, Boston, United States; [6]Department of Computational and Systems Biology, University of Pittsburgh School of Medicine, Pittsburgh, United States; [7]Department of Biochemistry and Biophysics, University of Pennsylvania, Penn Center for Genome Integrity, Philadelphia, United States; [8]Department of Obstetrics, Gynecology and Reproductive Sciences, University of Pittsburgh School of Medicine, Pittsburgh, United States

*For correspondence: yanowitzjl@mwri.magee.edu

Present address: †Lester and Sue Smith Breast Center, Department of Molecular and Cellular Biology, Baylor College of Medicine, Houston, United States

Competing interest: The authors declare that no competing interests exist.

## eLife Assessment

This study combines genetic, cell biological, and interaction data to propose a model of meiotic double-strand break regulation in *C. elegans*. **Solid** evidence supports the main conclusions, while by nature of a screening-type study, more may be needed to solidify speculations in future studies. Yet, comprehensive cataloging of the physical and genetic interactions of factors required for meiotic double-strand break is **useful** information for the field.

**Abstract** Double-strand breaks (DSBs) are the most deleterious lesions experienced by our genome. Yet, DSBs are intentionally induced during gamete formation to promote the exchange of genetic material between homologous chromosomes. While the conserved topoisomerase-like enzyme Spo11 catalyzes DSBs, additional regulatory proteins—referred to as 'Spo11 accessory factors'—regulate the number, timing, and placement of DSBs during meiotic prophase, ensuring that SPO-11 does not wreak havoc on the genome. Despite the importance of the accessory factors, they are poorly conserved at the sequence level, suggesting that these factors may adopt unique functions in different species. In this work, we present a detailed analysis of the genetic and physical interactions between the DSB factors in the nematode *Caenorhabditis elegans*, providing new insights into conserved and novel functions of these proteins. This work shows that HIM-5 is the determinant of X-chromosome-specific crossovers and that its retention in the nucleus is dependent on DSB-1, the sole accessory factor that interacts with SPO-11. We further provide evidence that HIM-5 mediates interactions with the different accessory factors subgroups, providing insights into how components on the DNA loops may interact with the chromosome axis.

## Introduction

One of the seminal events during meiosis is the formation of crossovers (CO) during the first meiotic division. While COs create new combinations of alleles through the exchange of genetic material between homologs, they also establish physical connections that ensure their proper alignment on the meiotic spindle and their subsequent segregation to opposite ends of the spindle. In the absence of COs, homologs segregate randomly, leading to gametes with altered chromosome numbers and subsequent fetal aneuploidies. In humans, aneuploidy is observed in 1 of 160 live births (*Dungan, 2002*) and in >50% of miscarriages (*Hassold et al., 2007*) underscoring the importance of proper CO formation.

A necessary early step in CO formation is the creation of DNA double-strand breaks (DSBs) catalyzed by the widely conserved Spo11 enzyme (*Bergerat et al., 1997*; *Keeney et al., 1997*). Although Spo11 is essential for DSB formation, it does not function alone: additional proteins regulate the timing, placement, and number of DSBs (*Cole et al., 2010*; *Hinman et al., 2021*; *Keeney, 2008*; *Vrielynck et al., 2021*). In *Saccharomyces cerevisiae*, where DSB formation has been best characterized, at least nine other proteins interact with Spo11 to regulate its recruitment and activation (*Lam and Keeney, 2015*; *Martini et al., 2006*). In vivo and in vitro physical interaction studies have revealed that these proteins form several subcomplexes. Briefly, the RMM complex (Rec114, Mer2, and Mei4) is loaded on chromosomes early in prophase and is required for Spo11 binding to sites of DNA cleavage (*Li et al., 2006*; *Sasanuma et al., 2007*). Activation of Mer2 requires phosphorylation by cyclin-dependent kinase (CDK) (*Henderson et al., 2006*), suggesting that Mer2 links DSB formation to the progression of the meiotic program (*Arora et al., 2004*; *Henderson et al., 2006*; *Li et al., 2006*; *Maleki et al., 2007*). Rec102 and Rec104 form another subcomplex that facilitates Spo11 dimerization and its association with DSB sites (*Arora et al., 2004*; *Kee and Keeney, 2002*; *Prieler et al., 2005*; *Sasanuma et al., 2007*). Ski8 binds Spo11 and is important for Spo11 nuclear localization (*Arora et al., 2004*). Finally, the Mre11–Rad50–Xrs2 (MRX) complex is also required for DSB formation but requires all the other DSB proteins to associate with DSB sites (*Bouuaert et al., 2021*). Despite their importance, most of these proteins are poorly conserved at the amino acid level, although homologs have been identified for several of these in organisms ranging from plants to worms to mice (*Cole et al., 2010*; *Kumar and De Massy, 2010b*). Importantly, a recent study of plants has presented a similar model for DSB induction as in yeast, yet the specific players and details differ (*Vrielynck et al., 2021*), highlighting the value of comparative studies to understand the evolutionary history of the DSB machinery in metazoans.

In vitro biochemical studies have shown that *Caenorhabditis elegans* SPO-11 is monomeric but that even when it is able to bind dsDNA, it does not exhibit DNA cleavage activity (*Yeh et al., 2017*). This observation strongly supports the hypothesis that cofactors are needed for DSB induction. DSB-1 and DSB-2 are homologs of the conserved Rec114 protein (*Tessé et al., 2017*), and DSB-3, a homolog of Mei4 (*Hinman et al., 2021*). These factors localize to chromosomes during early meiotic prophase and promote the activity of the SPO-11 complex (*Rosu et al., 2013*; *Stamper et al., 2013*; *Hinman et al., 2021*), functioning analogously to their counterparts in other species (*Kumar et al., 2010a*; *Kumar et al., 2015*; *Kumar et al., 2018*; *Li et al., 2006*; *Maleki et al., 2007*). *him-17*, *mre-11*, and *rad-50* are also essential for the introduction of breaks in this organism (*Chin and Villeneuve, 2001*; *Hayashi et al., 2007*; *Hinman et al., 2021*; *Reddy and Villeneuve, 2004*; *Stamper et al., 2013*).

Other genes, including *xnd-1*, *him-5*, *rec-1*, *cep-1*, and *parg-1*, have more limited roles in break induction. *him-5* is fascinating in that it is required to ensure the complement of DSBs on the X chromosome (*Janisiw et al., 2020*; *Meneely et al., 2012*; *Rosu et al., 2013*). In the region of the germ line where DSBs are made, the X chromosome is transcriptionally quiescent and packaged in a heterochromatin-like state (*Fong et al., 2002*; *Kelly et al., 2002*). By contrast, the autosomes are transcriptionally active and replete with histone modifications associated with open chromatin. It is thought that the heterochromatic-like state of the X presents a barrier to DSB formation necessitating the evolution of proteins to overcome this obstacle (*Wagner et al., 2010*). *xnd-1* transcriptionally regulates *him-5* to promote breaks on the X (*McClendon et al., 2016*), but it also restricts histone acetylation, a function that ensures the proper timing of DSB induction (*Wagner et al., 2010*). In the absence of *xnd-1*, there is precocious formation of DSBs as monitored by the accumulation of the HR strand-exchange protein RAD-51 in transition zone nuclei (*Gao et al., 2015*; *McClendon et al., 2016*; *Wagner et al., 2010*).

CO positioning is altered in all situations where DSB levels are affected, but the *rec-1* mutation has a profound effect on the distribution of meiotic COs (*Rose and Baillie, 1979*) with only a minor reduction in total DSB numbers (*Chung et al., 2015*). Prior analysis of *rec-1* and *him-5* found these genes to be paralogs that function cooperatively to ensure wild-type DSB levels (*Chung et al., 2015*). Interestingly, while loss of *rec-1* preferentially affects CO distribution, loss of *him-5* reduces DSB numbers nearly in half, severely delays DSB induction (until mid-late pachytene), and changes CO distribution (*Meneely et al., 2012*). *him-5* has redundant functions with *cep-1* for both break formation and inhibition of non-homologous end joining (NHEJ) (*Mateo et al., 2016*), revealing roles for *him-5* in coordinating DSB induction with downstream repair.

In this study, we expand the analysis of the genetic interactions between the SPO-11 accessory factors in *C. elegans* and provide the first protein–protein interaction studies of these factors using co-immunoprecipitation and yeast two-hybrid (Y2H). We use these interaction data to propose how the DSB formation is initiated in *C. elegans* and compare and contrast this to models proposed in yeast and plants (*Bouuaert et al., 2021*; *Vrielynck et al., 2021*). Our results allow us to begin to elucidate the regulatory events that control the localization, timing, and placement of meiotic DSBs in *C. elegans*.

## Results

### HIM-5 is a limiting factor for the HIM-17/XND-1-dependent breaks on the X chromosome

Three of the DSB proteins appear to have a biased effect on X chromosome CO formation: null alleles of *xnd-1* and *him-5* predominantly show a loss of CO intermediates on the X, increases in univalent X chromosomes at diakinesis in half or almost all nuclei, respectively, and an increase in XO male offspring resulting from nondisjunction of the X chromosome during the meiotic division; the hypomorphic *him-17(e2806)* mutation shows increased incidence of one unattached homolog pair in diakinesis and an increase in male production consistent with an X-specific defect (*Reddy and Villeneuve, 2004*). To determine if these genes function through a common underlying mechanism, we first performed Y2H assessing interactions with both high and low stringency. We observed a strong Y2H interaction between XND-1 and HIM-17, and a weak one between HIM-17 and HIM-5 (*Figure 1A*, *Figure 1— figure supplement 1*). To validate these interactions, we also performed immunoprecipitations (IPs) of HIM-17::GFP followed by mass spectrometry (MS, *Figure 1—figure supplement 1B*). This identified XND-1, as well as 16 additional strong interactors of HIM-17::GFP (*Figure 1—figure supplement 2*). Moreover, endogenous XND-1 was identified in western blots of IPs performed independently using 3xHA-tagged HIM-17 epitopes, indicating a robust interaction between these proteins (*Figure 1B*, *Figure 1—figure supplement 1C*). In contrast, HIM-5 was not detected in either the HIM-17::GFP co-IPs or IP/MS, suggesting that the interaction observed by Y2H either exists very transiently or was bridged by a protein expressed in yeast.

Prior studies from our group indicate that *xnd-1* regulates HIM-5 transcriptionally since ectopic expression of *him-5::GFP* using the *pie-1* germline promoter, but not expression using the endogenous *him-5* promoter, rescues the X chromosome nondisjunction defects conferred by the *xnd-1* mutation (*Wagner et al., 2010*; *Meneely et al., 2012*; *McClendon et al., 2016*). Since RNA-Seq studies showed that *him-17* regulates many germline genes including *him-5* (*Carelli et al., 2022*), we set out to determine if ectopic expression of *him-5* could also rescue *him-17* null mutants. Similar to what we observed with *xnd-1* mutants, *Ppie-1::him-5*—but not *Phim-5::him-5*—rescued the X chromosome defects seen by the increase of XO male offspring from XX hermaphrodites (known as the high incidence of males or 'Him' phenotype) (35% of males in *him-17(ok424)* vs. 7.02% in *Ppie1::him-5;him-17(ok424)*) (*Figure 1C*). Consistent with the lack of rescue, there was no statistically significant change in univalent formation in *Phim-5::him-5;him-17(ok424)* (*Figure 1—figure supplement 3*). These results cannot be explained by HIM-5 protein levels which are lower in *Ppie-1::him-5* than in *Phim-5::him-5* (*McClendon et al., 2016*). The lack of rescue from *Phim-5::him-5* supports the conclusion that HIM-17 directly regulates *him-5* expression. Since the *pie-1* promoter appears to be independent of HIM-17, the *Ppie-1::him-5* construct can—at least partially—bypass this regulation and suppress the *him-17* mutant phenotype. These findings suggest that both XND-1 and HIM-17 promote X chromosome DSB and CO formation by regulating *him-5* expression. Moreover, *xnd-1* and *him-17* have distinct

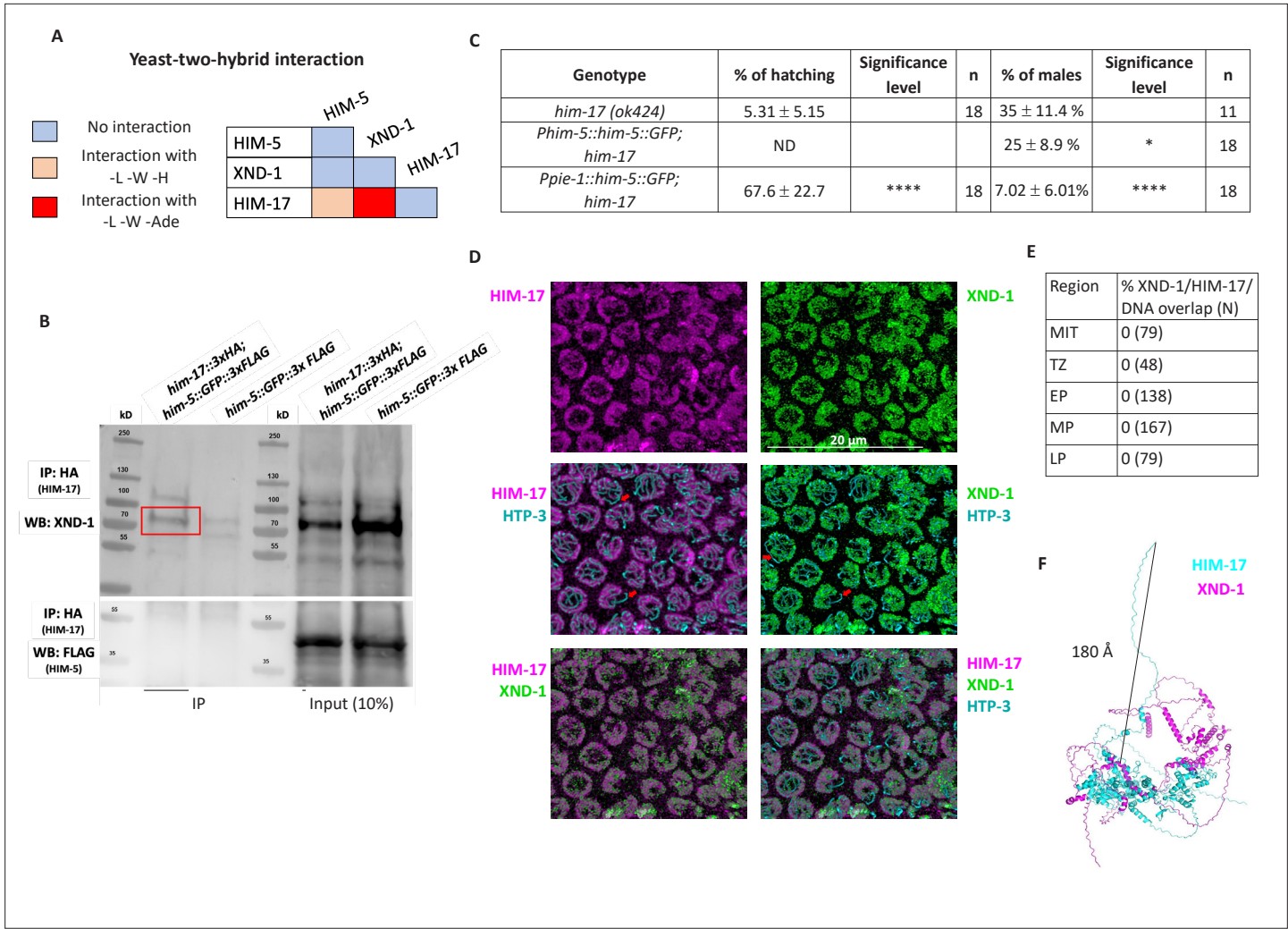

**Figure 1.** XND-1 and HIM-17 interact and are predominantly localized on DNA loops. (**A**) Summary of the yeast two-hybrid assay results. The complete set of results is given in *Figure 1—figure supplement 1A*. (**B**) Western blot analysis of endogenous XND-1 (top) and HIM-5::GFP::FLAG (bottom) on HA pulldowns performed in *him-17::3xHA; him-5::GFP::3xFLAG* strain. The red box indicates the XND-1 band. Analysis was performed in biological duplicates. Specificity of the anti-XND-1 antibody is shown in *Figure 1—figure supplement 4A*. (**C**) Quantification of embryonic viability (hatching rates) and frequencies of male offspring among the progeny of the indicated genotypes. Data are shown as mean ± SD; *p < 0.05, ****p < 0.0001, ND = not determined, $n$ = number of $P_0$ parents whose brood was examined. (**D**) Representative confocal images of mid-pachytene stage nuclei showing that juxtaposed XND-1 and HIM-17 localization away from the chromosome axes (stained with anti-HTP-3). The X chromosome is not stained by either XND-1 and HIM-17 (red arrows). Scale bars = 20 μm. (**E**) Quantification of the pairwise overlap between XND-1, HIM-17, and the chromosome axes. (**F**) Alphafold3 model of XND-1–HIM-17 dimer. The black line shows the 180 Å distance between the HA tag in HIM-17 and the XND-1 protein which is recognized by the polyclonal antibody.

The online version of this article includes the following source data and figure supplement(s) for figure 1:

**Source data 1.** PDF file containing original western blots for *Figure 1B* (top).

**Source data 2.** PDF file containing original western blots for *Figure 1B* (top), indicating the relevant bands.

**Source data 3.** PDF file containing original western blots for *Figure 1B* (bottom).

**Source data 4.** PDF file containing original western blots for *Figure 1B* (bottom), indicating the relevant bands.

**Source data 5.** Excel file containing row data used for quantification in *Figure 1C*.

**Figure supplement 1.** Yeast two-hybrid (Y2H) studies and control immunoprecipitation (IP) data.

**Figure supplement 1—source data 1.** PDF file containing original western blots for *Figure 1—figure supplement 1B*.

**Figure supplement 1—source data 2.** PDF file containing original western blots for *Figure 1—figure supplement 1B*, indicating the relevant bands.

**Figure supplement 1—source data 3.** PDF file containing original western blots for *Figure 1—figure supplement 1C*.

*Figure 1 continued on next page*

*Figure 1 continued*

**Figure supplement 1—source data 4.** PDF file containing original western blots for *Figure 1—figure supplement 1C*, indicating the relevant bands.

**Figure supplement 2.** Immunoprecipitation and mass spectrometry (IP-MS) results of HIM-17::GFP from whole worm extracts.

**Figure supplement 3.** Quantification of DAPI-stained bodies at diakinesis in *him-17(ok424)* shows lack of rescue by HIM-5 expressed from its endogenous promoter.

**Figure supplement 4.** Localization of XND-1 and HIM-17 is non-overlapping and not interdependent.

roles in DSB/CO formation since only null alleles of *him-17*, but not *xnd-1* or *him-5*, confer a severe loss of DSBs (*Meneely et al., 2012*; *Reddy and Villeneuve, 2004*; *Wagner et al., 2010*), while only *xnd-1* mutations influence the timing of DSB induction/repair (*Wagner et al., 2010*; *McClendon et al., 2016*).

We note that ectopic HIM-5 expression strongly rescued the embryonic lethality conferred by the *him-17* null allele (*Figure 1C*). While a small fraction of the lethality is attributed to X chromosome nondisjunction, most of the lethality is a consequence of autosomal missegregation. Thus, even though *him-5* null mutant animals exhibit a profound loss of DSBs/COs on the X chromosomes with rare CO defects on autosomes (*Meneely et al., 2012*), *him-5* expression can restore CO formation to both autosomes and X chromosomes. This strongly supports a general role for HIM-5 in CO formation and further indicates that HIM-5 is sufficient to restore COs on the X chromosome.

To further characterize the interaction between XND-1 and HIM-17, we also sought to determine whether they colocalize on meiotic chromosomes using confocal microscopy. Prior studies have shown that HIM-17 associates with meiotic chromatin (*Reddy and Villeneuve, 2004*). Co-staining the HIM-17::HA transgenic line with anti-HA and anti-XND-1 antibodies showed that HIM-17, like XND-1, is enriched on autosomes (*Figure 1D*; *Wagner et al., 2010*). Both proteins showed no overlap with the chromosome axes (marked with HTP-3; *Figure 1D*) or 4',6-diamidino-2-phenylindole (DAPI; *Figure 1—figure supplement 4B*), although we cannot rule out that a small fraction of the population, either below the level of detection or transiently, localizes to the axes. These immunolocalization studies also show a strong juxtaposition of HIM-17 and XND-1, with no detectable overlap between their staining patterns in germ lines from the mitotic region through late pachytene, when XND-1 disappears from the nucleus (*Figure 1E*, *Figure 1—figure supplement 4B*). Structural modeling of the XND-1 and HIM-17 interaction (*Abramson et al., 2024*; *Comeau et al., 2004a*) provides a potential explanation for this observation: HIM-17 is a very large protein, and the HA-tag positioned at its amino terminus is predicted to be located at least 180 Å away from the protein:protein interaction interface (*Figure 1F*).

## DSB-1 mediates interactions with SPO-11 and promotes HIM-5 nuclear localization

Since HIM-5 appears to be central to DSB/CO formation on the X chromosome and the presence of DSB-1 and DSB-2 is required for all meiotic breaks, we next sought to determine whether HIM-5 interacts with DSB-1, its paralog DSB-2, or DSB-3. We found that DSB-1 and DSB-2 show robust Y2H interactions and that DSB-1 and DSB-3 exhibit weak Y2H interactions (*Figure 2A*, *Figure 2—figure supplement 1A*), confirming results from a prior study (*Hinman et al., 2021*). We also found that HIM-5 shows weak Y2H interactions with DSB-1 (*Figure 2A*). We confirmed the interaction between DSB-1 and HIM-5 with co-IP, pulling down GFP::DSB-1 and probing for HIM-5::3xHA by western blotting (*Figure 2B*).

We next wanted to understand the functional relevance of the interaction between DSB-1 and HIM-5. To this end, we examined the localization of HIM-5 proteins in the *dsb-1* mutant background. Co-staining of the chromosome axis (anti-HTP-3) and either HIM-5::GFP transgene (anti-GFP) or endogenously tagged HIM-5::3xHA (anti-HA) revealed that HIM-5 nuclear localization begins just prior to the transition zone in both wild-type and *dsb-1* mutants (*Figure 2C*, *Figure 2—figure supplement 1A*, arrows). However, whereas in wild-type, HIM-5 protein remains nuclear throughout early-mid pachytene and becomes further enriched on meiotic chromatin; in *dsb-1* mutants, HIM-5 nuclear localization is lost (*Figure 2C*, *Figure 2—figure supplement 1B*). From this change in HIM-5 protein distribution, we infer that DSB-1 (and/or its DSB-promoting function) is required to retain HIM-5 in meiotic nuclei.

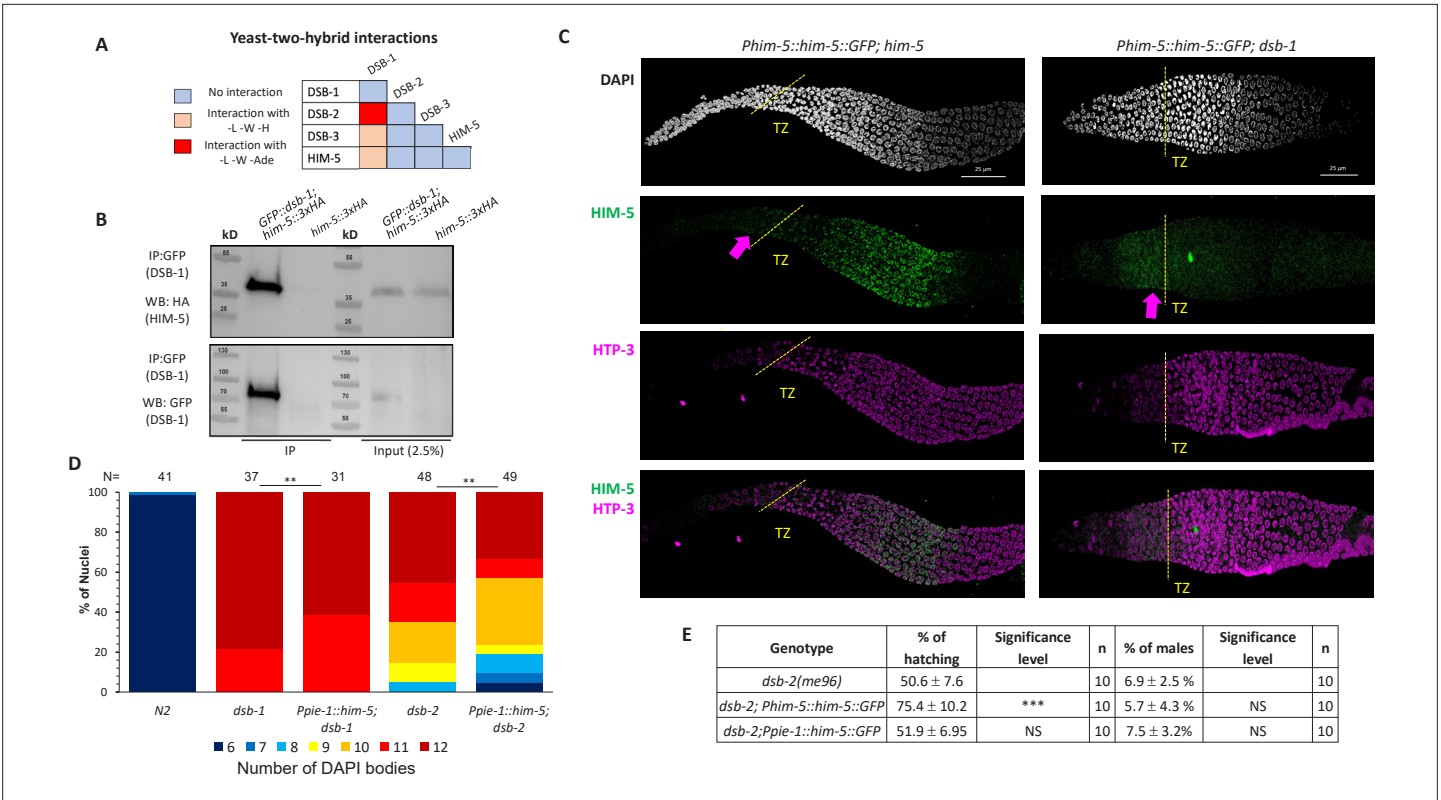

**Figure 2.** DSB-1 associates with HIM-5 and regulates its localization. (**A**) Summary of yeast two-hybrid results with *dsb-1*, *dsb-2*, *dsb-3*, and *him-5*. The complete set of results is given in **Figure 1—figure supplement 1A**. (**B**) Western blot analysis of anti-GFP pulldowns performed in *GFP::dsb-1;him-5::3XHA* strain showing co-IP of HA-tagged HIM-5 proteins. Analysis was performed in biological duplicates. (**C**) Immunofluorescence analysis of HIM-5 localization in *Phim-5::him-5::GFP;him-5* and *Phim-5::him-5::GFP;dsb-1* backgrounds. DNA (DAPI, gray), HIM-5::GFP (anti-GFP, green), and chromosome axes (anti-pHTP-3, magenta). pHTP-3 staining delineates entrance to the transition zone (marked dotted yellow lines), while arrows indicate the pre-meiotic localization of HIM-5. (**D**) Quantification of DAPI-stained bodies at diakinesis for the indicated genotypes. Colors correspond to the number of DAPI-stained bodies shown in the key below for *N2*, *dsb-1*, and *dsb-2* mutants expressing *Ppie-1::him-5::GFP*. Sample sizes (*N*) are indicated. Statistical significance for comparisons between groups is shown at the top (**p < 0.01). (**E**) Quantification of embryonic viability (hatching rates) and frequencies of male offspring among the progeny of the indicated genotypes. Data are shown as mean ± SD; NS stands for not significant; ***p < 0.001, *n* = number of $P_0$ parents whose brood was examined.

The online version of this article includes the following source data and figure supplement(s) for figure 2:

**Source data 1.** PDF file containing original western blots for **Figure 2B** (top).

**Source data 2.** PDF file containing original western blots for **Figure 2B** (top), indicating the relevant bands.

**Source data 3.** PDF file containing original western blots for **Figure 2B** (bottom).

**Source data 4.** PDF file containing original western blots for **Figure 2B** (bottom), indicating the relevant bands.

**Source data 5.** Excel file containing row data used for quantification in **Figure 2D**.

**Source data 6.** Excel file containing row data used for quantification in **Figure 2E**.

**Figure supplement 1.** Localization of HIM-5 in *dsb-1* mutants.

Since HIM-5 levels could also be affected by *dsb-1* and/or *dsb-2*, we next asked whether ectopic expression of HIM-5 could suppress defects in bivalent formation due to the loss of *dsb-1* or *dsb-2*. As shown in **Figure 2D**, ectopic HIM-5 reduced the number of univalents in both mutant backgrounds. Since a small fraction of *dsb-2; Ppie-1::him-5* animals exhibited six or seven DAPI bodies, we assayed whether the transgene could improve hatching rates and reduce male production in *dsb-2* (**Figure 2E**). Whereas the *Ppie-1::him-5* transgene has no impact on hatching and male frequencies, the more highly expressed *Phim-5::him-5* transgene (**McClendon et al., 2016**) conferred a mild suppression of the embryonic lethality. This ability to suppress hatching—but not male production—further supports a role for HIM-5 in regulating total DSB numbers.

## Genetic interactions define four functional groups that contribute to DSB/CO formation

Having established interactions between subsets of the SPO-11 accessory factors in *C. elegans*, we next wanted to perform more extensive genetic analyses between the known DSB factors. To this end, we created double mutant combinations of alleles with partial DSB defects. These included either weak loss-of-function alleles of essential DSB genes (*him-17* and *mre-11*) or null alleles of genes that partially reduce DSBs (*cep-1*, *dsb-2*, *rec-1*, *him-5*, and *parg-1*). Diakinesis oocytes were analyzed in the single and double mutants by whole-mount fixation and DAPI staining followed by confocal microscopy and 3D visualization. The chromosomes are highly condensed at diakinesis, and six bivalents can be detected in wild-type worms. Univalents, fusions, and DNA fragments can be seen in mutants with defects in DSB formation or their repair into a CO, respectively (*Chin and Villeneuve, 2001*; *Guo et al., 2022*; *Rosu et al., 2013*). We hypothesized that double mutant strains carrying mutations in genes that function in different aspects of DSB induction would show more severe defects and have an increase in univalent formation compared to single mutants. On the other hand, genes that function together or at the same step of break induction should exhibit no or only mild enhancement of the DSB defects.

Double mutant analysis was precluded with *dsb-1* and *dsb-3* since the single mutants already show a complete absence of breaks (*Hinman et al., 2021*; *Stamper et al., 2013*). By contrast, *dsb-2* mutant nuclei manifest an age-dependent loss in DSB capacity (*Rosu et al., 2013*; *Stamper et al., 2013*) allowing us to analyze interactions between *dsb-2* and *cep-1*, *parg-1*, *him-17*, and *rec-1* (*Figure 3A*, *Figure 3—figure supplement 1A–D*). As 1-day-old adults, *dsb-2* single mutants had ~50% of diakinesis nuclei that contained 6 bivalents, ~30% with 7 DAPI-stained bodies (5 bivalents and 2 univalents), and ~20% with 8 or more DAPI-stained bodies. Single mutants of *cep-1*, *parg-1*, and *rec-1* each had >90% of diakinesis nuclei with 6 bivalents, consistent with prior studies (*Chung et al., 2015*; *Janisiw et al., 2020*; *Mateo et al., 2016*). In animals carrying *him-17(e2806)* mutation, a weak allele of the otherwise essential locus for DSB formation, only ~50% of mutant nuclei had six bivalents; the other 50% had seven DAPI-stained bodies, reflecting the requirement for this gene in CO formation on the X chromosome (*Reddy and Villeneuve, 2004*). When *dsb-2* was combined with each of these mutations, an increased number of DAPI-stained bodies and a concomitant decrease in the number of chiasmata were observed (*Figure 3A*, *Figure 3—figure supplement 1*). These results strongly suggest that *dsb-2* functions cooperatively with all four loci for the regulation of meiotic CO formation.

Given that *rec-1*, *parg-1*, and *cep-1* fall in a separate functional group from *dsb-2*, we next addressed whether these three genes function together or independently. All double mutant permutations of *rec-1*, *parg-1*, and *cep-1* were analyzed, and in each case, the single and double mutants were nearly indistinguishable from the single mutants, that is, six bivalents (*Figure 3—figure supplement 1L, M*). Therefore, we infer that these genes likely function to control the same step in DSB/CO induction (*Figure 3B*). Consistent with this interpretation, when *rec-1*, *parg-1*, or *cep-1* were combined with other DSB gene mutations, including *him-17*, they had similar effects, and the numbers of univalents were increased (*Figure 3—figure supplement 1A, C, D* for Group 1; *Figure 3—figure supplement 1E–G* for Group 3; and *Figure 3—figure supplement 1J, K* for Group 4). Similar results have previously been reported from our laboratories for double mutants with *him-5* (*Chung et al., 2015*; *Janisiw et al., 2020*; *Mateo et al., 2016*).

*mre-11* functions in both DSB formation and subsequent DNA end resection (*Chin and Villeneuve, 2001*), the latter as part of the MRE11–RAD50–NBS1/Xrs2 (MRN/X) resection complex, the other proteins of which are encoded by the worm *rad-50* and *nbs-1* genes (*Girard et al., 2018*; *Hayashi et al., 2007*). Null mutations in *mre-11* and *rad-50* completely abrogate DSB induction. However, we were able to take advantage of a separation-of-function allele, *mre-11(iow1)*, that cannot perform its end-resection functions but can induce DSBs (*Yin and Smolikove, 2013*). The *iow1* mutation causes chromosome fusions to appear in diakinesis oocytes resulting from the aggregation of unprocessed DNA ends. While this allele makes enough DSBs to cause aggregation of all chromosomes, we hypothesized that it might have a minor impairment in DSB induction that would be revealed when combined with other DSB mutations. If this were the case, the double mutants would make fewer breaks, leading to fewer chromosome fusions and an increase in the number of DAPI-stained bodies/univalents. Consistent with this hypothesis, *mre-11(iow1)* double mutants with each of the other DSB mutations led to fewer fusions and more DAPI-stained bodies (*Figure 3A*, *Figure 3—figure supplement 1H–K*).

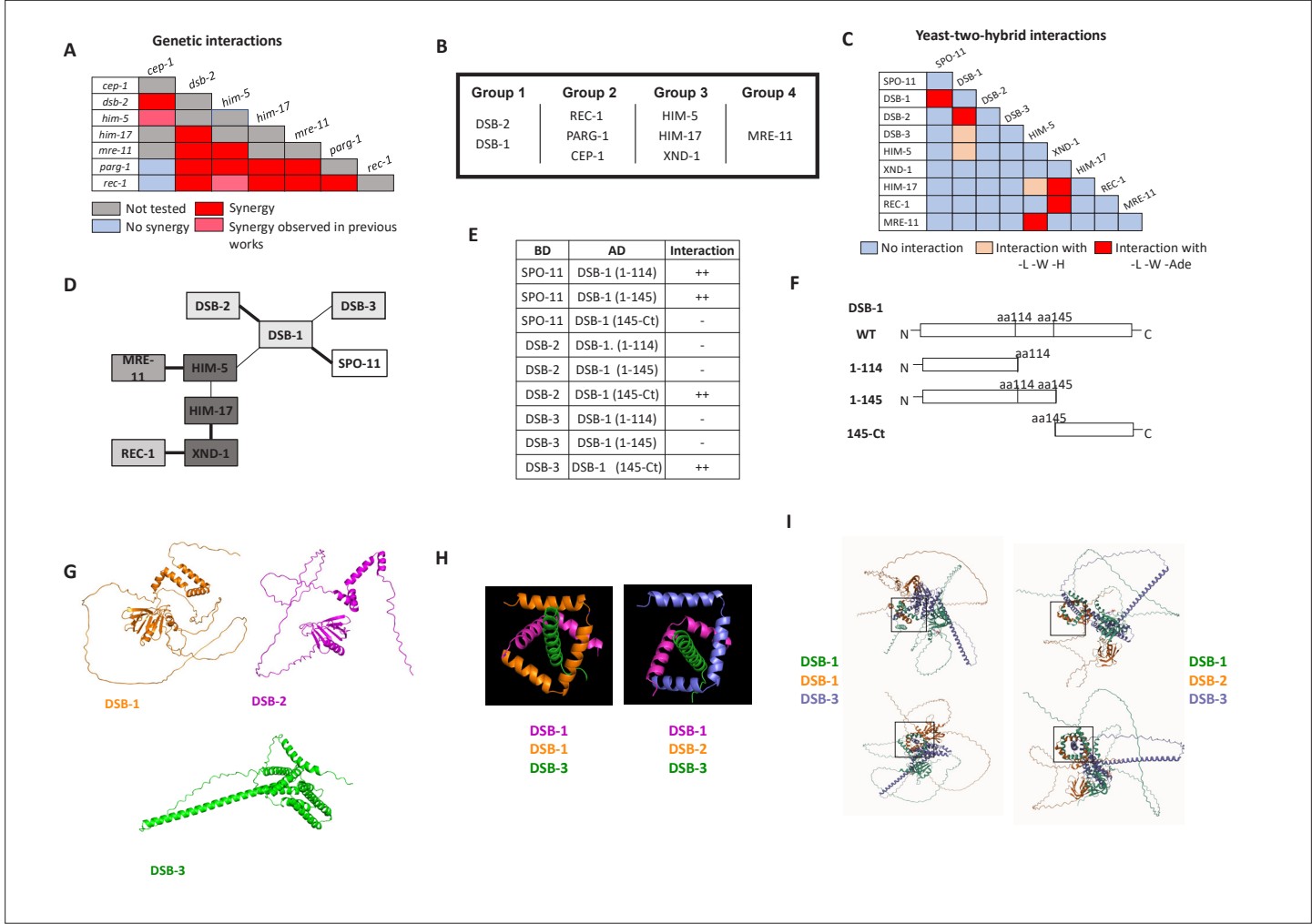

**Figure 3.** Genetic and protein–protein interaction studies show evidence of SPO-11 accessory protein sub-complexes in *C. elegans*. (**A**) Summary of the genetic interaction results. The complete set of results is given in *Figure 3—figure supplement 1*. (**B**) Genetic groupings based on data presented here and previously published (*Chung et al., 2015*; *Mateo et al., 2016*; *McClendon et al., 2016*; *Janisiw et al., 2020*). (**C**) Summary of yeast two-hybrid (Y2H) results. The complete set of results is shown in *Figure 1—figure supplement 1A*. (**D**) Schematic representation of the interaction network based on Y2H interactions. Thick line: Strong interaction; thin line: weak interaction. (**E, F**) Y2H interactions of SPO-11, DSB-2, and DSB-3 with different DSB-1 sub-domains (schematic in **F**). Details on DSB-1 sequence, protein structure, and deletions used in these experiments are found in *Figure 3—figure supplement 4*. (**G**) AlphaFold2 predictions of DSB-1, DSB-2, and DSB-3 protein structures. (**H, I**) Predicted interaction domains of DSB-1, DSB-2 with DSB-3 based on homology with the yeast proteins. The DSB-1/-1/-3 trimer is thermodynamically more stable than the DSB-1/-2/-3 trimer, consistent with the subordinate role for DSB-2 in break induction in young animals.

The online version of this article includes the following source data and figure supplement(s) for figure 3:

**Figure supplement 1.** Epistasis analysis of DSB factors defines multiple genetic groups for crossover formation.

**Figure supplement 1—source data 1.** Excel file containing row data used for quantification in *Figure 3—figure supplement 1*.

**Figure supplement 2.** Mixed phenotypes are seen in *cep-1(lg12501);mre-11(iow1)* double mutants.

**Figure supplement 2—source data 1.** Excel file containing row data used for quantification in *Figure 3—figure supplement 2A and C*.

**Figure supplement 3.** Irradiation rescues crossover defects of accessory factor double mutant strains.

**Figure supplement 3—source data 1.** Excel file containing row data used for quantification in *Figure 3—figure supplement 3*.

**Figure supplement 4.** Structural analysis of DSB-1 identified potential interaction motifs.

Thus, we propose that *mre-11*, and by extension *rad-50*, function in their own branch of the DSB/CO pathway.

Surprisingly, a unique phenotype was observed in a subset of the *cep-1; mre-11* double mutants. Our prior studies showed that *cep-1* functions redundantly with *him-5* to both ensure DSB formation

and promote downstream homologous recombination (HR) repair by preventing NHEJ (*Mateo et al., 2016*). In that study, we reported that *cep-1(lg12501)* is a separation-of-function allele that is defective in DSB formation. In the *cep-1(lg12501); mre-11(iow-1)* double mutants, we observed mixed pheno-types: some nuclei had 12 DAPI-stained bodies, while others exhibited an exacerbation of the fusion defect (*Figure 3—figure supplement 2*). We infer that the univalents indicate a synergistic effect between *mre-11* and *cep-1* for DSB formation; the increase in fusions, on the other hand, may suggest that *cep-1(lg12501)* is not fully wild-type for the DNA repair functions of CEP-1.

While each of the genes that we analyzed has reported roles in DSB formation, we wanted to confirm that the observed increases in univalents in the double mutants were in fact due to impairment in DSB induction and not to deficits in later steps of CO formation. Therefore, we relied on an established assay to distinguish between these possibilities: we asked whether the addition of exogenous breaks from gamma irradiation (IR) could suppress the univalent phenotype by serving as a surrogate for SPO-11-induced DSBs (*Dernburg et al., 1998*). Recent work has shown that 10 Gy IR is able to produce up to ~20 DSBs per meiotic nucleus (*Lascarez-Lagunas et al., 2023*), which is sufficient to ensure a CO on each chromosome (*Mets and Meyer, 2009*). We analyzed the number of DAPI bodies with and without IR in a subset of the double mutants where we observed synergistic effects (*dsb-2; parg-1*, *dsb-2; rec-1*, and *rec-1; him-17*). When each of these double mutant strains was exposed to IR, the wild-type phenotype of 6 bivalents was largely restored (*Figure 3—figure supplement 3*), supporting the conclusion that the CO deficiency observed is triggered by defects in DSB formation.

## Protein–protein interactions between the DSB regulatory factors

The genetic analyses described above defined four functional groups that contribute to break formation (*Figure 3B*). Using Y2H, we next set out to determine whether interactions between and within groups could explain how the different complexes come together to regulate DSB induction (*Figure 3C, D*, *Figure 1—figure supplement 1A*). CEP-1 showed auto-activation in both vectors and therefore was removed from further analysis. DSB-1, HIM-17, and REC-1 also showed self-activation in the binding domain vector; therefore, only the activation domain fusions were analyzed. Pairwise interactions between all remaining combinations were then analyzed. HIM-5 displayed the most promiscuous Y2H interactions, with strong interactions seen with MRE-11 and weak interactions as already discussed with HIM-17 and DSB-1. XND-1, apart from interacting with HIM-17, also showed strong Y2H interactions with REC-1. A recent study provided Y2H evidence supporting the idea that SPO-11 interacts with DSB-1 (*Hinman et al., 2021*). Critically, our work supports and extends this observation to show that DSB-1 is the only known accessory factor that interacts with SPO-11 in Y2H assays, cementing its role as a key determinant for DSB induction and explaining its role in feedback controls (*Stamper et al., 2013*; *Hinman et al., 2021*; *Guo et al., 2022*).

To analyze more in-depth the Y2H interactions between DSB-1 and DSB-2, DSB-3, and SPO-11, we looked for protein domains in DSB-1 that could be involved in these interactions. Interestingly, DSB-1 has five 'sticky' helices (helices 2–6 in *Figure 3—figure supplement 4*) that have the potential to be implicated in protein–protein interactions. We first deleted all of these helices together by introducing a deletion after L114 (DSB-1 (1–114) in *Figure 3E, F*). We found that this truncated version of DSB-1 loses the ability to interact with DSB-2 and DSB-3 but is still able to associate with SPO-11. Moreover, when we only deleted the helices 3–6 but kept the helix 2 (DSB-1 (1–145)), we obtained the same results, suggesting that the Y2H interactions between DSB-1 and DSB-2/DSB-3 are mediated by the C-terminal helices 3–6 and not helix 2. To prove this, we created a truncated DSB-1 protein that only contains amino acids from P145 to the stop codon (DSB-1(145-Ct)). Consistent with the deletion analysis, we observed that the C-terminal domain is sufficient to promote associations with DSB-2 and DSB-3. This domain was also not sufficient to promote the DSB-1/SPO-11 interaction (*Figure 3E*). These experiments complement the results published by Villeneuve lab where they observed that the DSB-1 N-terminus is the domain necessary for the DSB-1/SPO-11 (but not for the DSB-1/2/3) interaction (*Hinman et al., 2021*).

We next used AlphaFold and structural modeling to investigate whether the *C. elegans* REC114/MEI4 homolog could assemble in the same fashion as their distant orthologues (*Claeys Bouuaert et al., 2021*; *Daccache et al., 2023*) in which two helices of REC114 dimer surround a MEI4 helix. DSB-1 is predicted to form a stable homodimer that can assemble into a highly stable trimeric complex with DSB-3 threaded through the middle (*Figure 3G–I*) analogous to what is predicted for

Rec114-Mei4 (*Daccache et al., 2023*). By contrast, DSB-2 cannot form homodimers, although it can replace one of the DSB-1 monomers to form the DSB-1:DSB-2:DSB-3 ternary complex. Free energy calculations (*Champ and Camacho, 2007*) show that the DSB-1:DSB-2:DSB-3 complex is less stable (−3.8 kcal/mol) than a DSB-1:DSB-1:DSB-3 complex consistent with the more limited role for DSB-2 in break formation, at least in young worms (*Rosu et al., 2013*; *Stamper et al., 2013*).

## Discussion

Across species, accessory factors collaborate with SPO-11 to regulate various aspects of DSB formation, including the timing, placement, number, and persistence of DSB induction, as well as the chromatin architecture at and around the break site (*Arora et al., 2004*; *Cole et al., 2010*; *Hinman et al.,*

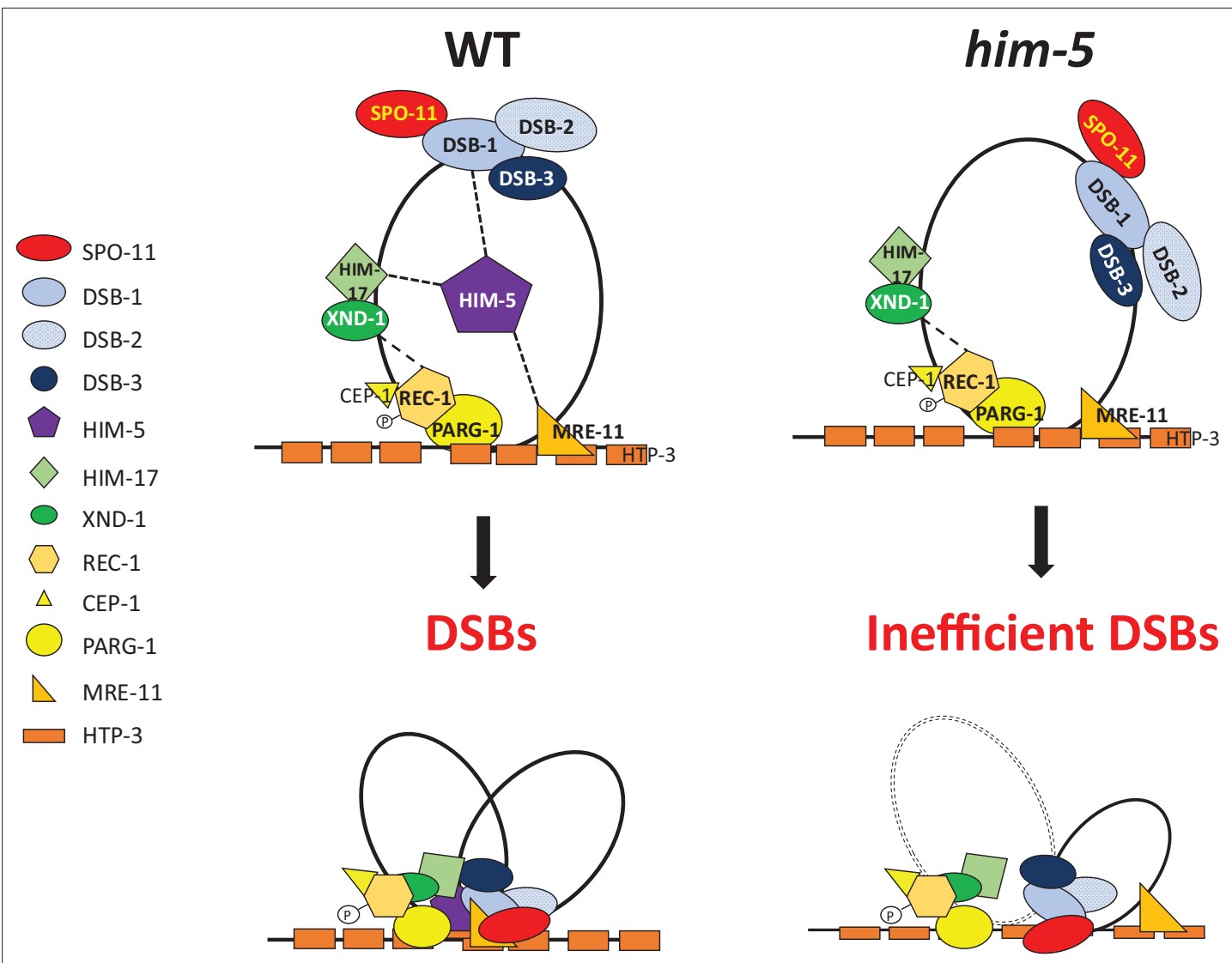

**Figure 4.** Speculative model of the DSB formation complex in *C. elegans*. Synthesizing prior work and data herein, we propose a model to explain the known interactions and localization patterns of the DSB regulatory factors in *C. elegans*. DSB-1/-2/-3 and HIM-17–XND-1 subcomplexes are initially found on DNA loops. The phosphorylation of REC-1 by CDK activates DSB formation, allowing HIM-5 through its association with MRE-11 and PARG-1 (which both associate with HTP-3) to efficiently bring the DSB-1/-2/-3 complex with SPO-11 to the chromosome axis. This recruitment to the axes by HIM-5 facilitates the coupling of DSBs to downstream repair by the MRN complex and others. In the absence of HIM-5, DSBs/COs on autosomes do not occur efficiently and crossover sites are shifted to the gene-rich third of each chromosome, presumably because DSB-1/-2/-3/SPO-11 cannot be as efficiently recruited to the proper sites partially defined by HIM-17 and XND-1. We assume that recruitment to the chromosome axes in the absence of HIM-5 is likely stochastic or mediated by as yet unknown protein interactions. On the X chromosome, the absence of HIM-5 prevents the formation of most DSBs.

*2021*; *Kumar and De Massy, 2010b*; *Panizza et al., 2011*; *Sasanuma et al., 2007*; *Sommermeyer et al., 2013*; *Vrielynck et al., 2021*). Here, we present genetic, biochemical, and cytological studies to develop a working interaction model for DSB regulation in *C. elegans*. The *C. elegans* DSB complex appears to be comprised of four sub-groups (*Figures 3B and 4*) interconnected by their association with HIM-5. Our results suggest two important roles for HIM-5: first as the linchpin between the whole DSB complex, thereby ensuring the coordination of the timing, placement, and number of DSBs (described in more detail below); and second, as the determinant in X chromosome COs.

DSB-1 and DSB-2 are both orthologs of the budding yeast Rec114 protein, but DSB-1 has a more central role in DSB induction (*Rosu et al., 2013*; *Stamper et al., 2013*), which we posit can be explained by the increased stability of the DSB-1:DSB-1:DSB-3 trimer compared to a DSB-1:DSB-2:DSB-3 trimer. The direct association between SPO-11 and DSB-1, but none of the other worm accessory factors, points to DSB-1/REC114 as the hub for the integration of break-inducing signals in this organism. In yeast and *Arabidopsis,* accessory proteins are associated indirectly with Spo11/SPO-11 through interactions with the TopVI/MTOPVIB subunit (*Arora et al., 2004*; *Kee and Keeney, 2002*; *Maleki et al., 2007*; *Robert et al., 2016*; *Vrielynck et al., 2021*). These differences between species underscore how conserved DSB proteins have been co-opted to regulate different aspects of break induction.

While DSB-1/-2/-3 are critical for CO formation, they predominantly localize on DNA loops with little to no overlap (*Hinman et al., 2021*). Instead, a small fraction of these three proteins appears to associate with the DNA axis, a localization that is consistent with the 'tethered loop model' for break induction in yeast and plants (*Kee et al., 2004*; *Maleki et al., 2007*; *Panizza et al., 2011*; *Sommermeyer et al., 2013*; *Tsai et al., 2020*). Two accessory factors, MRE-11 and PARG-1, are known interactors of the axis protein HTP-3 (*Janisiw et al., 2020*). Both proteins interact either directly or indirectly with HIM-5, which physically associates with DSB-1, thereby providing a mechanism to bring the loop-associated proteins to the axis (*Figure 4*).

MRE-11 and PARG-1 also have roles in both break formation and DNA repair. For PARG-1, we show here that it functions together with REC-1 and CEP-1 based on the following data: synthetic CO defects were seen in *cep-1;him-5*, *rec-1;him-5*, and *parg-1;him-5* (*Chung et al., 2015*; *Mateo et al., 2016* and this paper), but not in *rec-1;cep-1*, *rec-1;parg-1*, and *cep-1;parg-1* double mutants; *dsb-2* mutant phenotypes are exacerbated by the loss of *rec-1*, *cep-1* (*Figure 3—figure supplement 1*), and *parg*-1 (*Janisiw et al., 2020*). Interestingly, both CEP-1 and PARG-1 have additional roles in promoting HR-mediated repair and/or preventing alternative repair pathways (*Janisiw et al., 2020*; *Mateo et al., 2016*). We envision that CEP-1, REC-1, and PARG-1 function together to promote DSBs and their repair into COs at the chromosome axis. Since we also showed that the DSB defects in *mre-11* mutant animals were enhanced by loss of *parg-1* or *rec-1* (*Figure 3—figure supplement 1*), we favor a model in which MRE-11 and PARG-1 independently interact with the chromosome axis to ensure robust DSB formation and/or repair. It is possible that MRE-11, which is essential for COs, facilitates the tethering of chromosome loops to the axis, but further structural studies of these complexes are required.

The defects observed in *cep-1;him-5* (*Mateo et al., 2016*) and *rec-1;him-5* (*Chung et al., 2015*) are both exacerbated by maternal age, similar to *dsb-2*. We envision that inputs from both HIM-5 and DSB-2 converge on DSB-1 to ensure robust DSB formation across the lifespan. This would also explain the synthetic defects in DSB formation between *dsb-2* and each of the components in the *him-5* interaction network. DSB-1 is required for chromatin association of DSB-2 but not vice versa, although DSB-1 levels are reduced in older *dsb-2* mutant animals (*Rosu et al., 2013*; *Stamper et al., 2013*). We show here that DSB-1 is also required for nuclear retention of HIM-5. While the functional significance of this interaction awaits isolation of separation-of-function mutations that abrogate their interaction, it is tempting to speculate that the signals that initiate meiotic induction regulate this interaction.

## Timing of breaks

Studies from multiple systems have elucidated mechanisms to coordinate the timing of DSBs with the completion of meiosis, for example by CDK and Ddf4 phosphorylation in yeast (*Murakami and Keeney, 2008*; *Carelli et al., 2022*). In worms, REC-1 was previously shown to be a target of CDK, and therefore, it was hypothesized that such a phosphorylation event would be critical to ensure that break induction occurs after the completion of S phase (*Chung et al., 2015*). REC-1 and HIM-5 evolved from a gene duplication event in the *C. elegans* lineage, whereas the ancestral gene in related

nematodes has features of both proteins, although it shares greater homology with HIM-5 (*Chung et al., 2015*). Thus, we hypothesize that the single ancestral REC-1/HIM-5 protein would be subject to CDK phosphorylation, which would then promote the assembly of the complexes at the chromosome axis to promote break formation.

Timing also appears to be critical for CO outcomes. Studies in yeast show different outcomes for early and late breaks (*Joshi et al., 2015*), and recent studies in worms indicate that later breaks are needed for CO formation (*Hicks et al., 2022*). We previously showed that *xnd-1* mutants have altered DSB kinetics, showing elevated levels of DSB formation in leptotene/zygotene (*McClendon et al., 2016*; *Wagner et al., 2010*). This function of XND-1 is mediated, at least in part, by its effect on chromatin architecture. One possibility is that the physical interaction that we show here between XND-1 and REC-1 couples the completion of S phase to the activation of breaks within the correct chromatin environment, thus ensuring their timely formation. In the absence of XND-1, increases in global histone acetylation (*Gao et al., 2015*; *Wagner et al., 2010*) might allow SPO-11 easier access to the DNA. Another possibility is that upon phosphorylation of REC-1 by CDK-1, REC-1 through PARG-1 transiently brings XND-1 (the timer) to the axis, an event that restrains DSB formation.

## Chromatin and meiotic gene transcription

We provide strong evidence for direct interactions between XND-1 and HIM-17, both by Y2H and co-IP. HIM-17 appears to have a role in transcriptional regulation of DSB factors. Here, we show that ectopic expression of HIM-5 from the *pie-1* promoter, but not overexpression from the endogenous *him-5* promoter, suppressed *him-17* phenotypes. This is similar to the rescue we saw of *xnd-1* (*McClendon et al., 2016*) and suggests that both proteins contribute to the expression of *him-5*. This supports findings that *him-17* regulates the expression of ~300 germline-enriched genes, including *him-5*, *rec-1*, and *dsb-2* (*Carelli et al., 2022*), whose concomitant loss would be expected to abrogate break induction (*Chung et al., 2015* and *Figure 3A*, *Figure 3—figure supplement 1*). While its only role may be in transcriptional regulation, a growing literature and data presented herein suggest a more direct role for HIM-17 in meiosis. We presented Y2H data for a weak interaction between HIM-17 and HIM-5. Although these interactions were not confirmed by IP, it remains possible that these complexes are transient associations, are unstable during IP, and/or are not abundant. Together with observed changes in chromatin structure and both DSB and CO distribution in *him-17* mutants (*Nadarajan et al., 2021*; *Reddy and Villeneuve, 2004*), it is tempting to speculate that together with XND-1, HIM-17 helps to create an environment permissive for SPO-11 binding and/or break formation. XND-1 and HIM-17 may function together in this role or redundantly, perhaps through XND-1 interacting with REC-1 and HIM-17 interacting with HIM-5, or in related species through interactions with the single REC-1/HIM-5 paralog.

## X/autosome differences in DSB formation

One of the conundrums arising from our data is in the function of the HIM-5 protein. *him*-5 was identified based on its preferential impact on CO formation on the X chromosome (*Broverman and Meneely, 1994*; *Hodgkin et al., 1979*). The X-specific phenotypes conferred by the *him-5* mutation could suggest that the X chromosome is simply more sensitive to a reduction in the number of active SPO-11 complexes; the fact that mutations in *dsb-2* also reduce DSBs but without an X chromosome bias argues against this interpretation. We show herein that HIM-5 is sufficient to ensure that COs occur on the X chromosome. Ectopic expression of *him-5* can substantially rescue both the autosomal and X chromosomal defects of *him-17* null mutations—despite expression levels lower than endogenous *him-5* (*McClendon et al., 2016*). The X chromosome-specific defects of *him-5* mutations might arise due to defects in recruitment of DSB-1-2-3/SPO-11 to specific sites on the X or to an impaired ability to coordinate the assembly of the DSB machinery on the—late-replicating (*Jaramillo-Lambert et al., 2007*)—axis of the X chromosome, although other models can be envisioned.

Our data and a growing literature insinuate a more general and central role for HIM-5 in DSB formation and downstream CO repair: (1) *rec-1;him-5* double mutants give an age-dependent severe loss of DSBs (like *dsb-2* mutants) suggesting that the ancestral function of the protein may have a more profound effect on break formation (*Chung et al., 2015*); (2) total DSBs are reduced to nearly 50% of wild-type and are severely delayed in formation in *him-5* mutants (*Meneely et al., 2012*); (3) a recent study reports a role for *him-5* in CO formation in males where the X chromosome does not

engage in CO exchange (*Engebrecht et al., 2025*); (4) HIM-5 is required for the normal distribution of COs on the autosomes (*Meneely et al., 2012*); (5) redundantly with CEP-1, HIM-5 promotes DSB formation and HR-mediated repair and inhibits NHEJ (*Mateo et al., 2016*), and (6) our data herein show that HIM-5 interacts not only with DSB-1, but also components of the other functional groups that regulate DSB induction. Together, we believe this data supports a role for HIM-5 in bridging the accessory factors with SPO-11 to ensure the proper timing, placement, and number of breaks.

## Concluding remarks
Our Y2H and IP results show very strong protein–protein interactions between the different DSB factors; however, their localization along the gonads does not frequently = coincide suggesting temporal and spatial regulation throughout meiosis. Further studies are necessary to elucidate how these interactions arise and change in vivo and how these are integrated with both cell cycle and regulatory feedback controls.

## Methods
### Culture and strains
Worms were cultured on MyoB plates (*Burns et al., 2006*) seeded with OP50 at 20°C, unless otherwise noted (*Brenner, 1974*). Mutant strains used in this study are listed in Supplemental Material, *Supplementary file 1*. All strains were derived from the wild-type Bristol strain N2.

### Chromosome morphology analysis of diakinesis oocytes
The numbers of DAPI-stained bodies present in diakinesis oocytes were assessed in intact adult hermaphrodites at day 1 of adulthood (24 hr post-L4, unless otherwise specified). Adults were fixed in Carnoy's fixative solution (60% ethanol, 30% chloroform, and 10% glacial acetic acid) and stained with DAPI (10 mg/ml stock diluted 1:50,000 in 1X phosphate-buffered saline; PBS) for 15 min in a humid chamber. Worms were mounted in Prolong Gold with DAPI, cured overnight at room temperature, and stored at 4°C prior to imaging either on a Nikon A1r confocal microscope or Leica Stellaris 5 confocal with an integrated White Light Laser (WLL). Diakinesis images were procured as 0.2 μm per plain Z-stacks and visualized using Volocity 3-D Imaging Software for Nikon images and LAS X for Leica images. Fisher tests were performed to analyze the DAPI-stained chromosome morphology data in diakinesis oocytes. This approach was used to compare the distribution of DAPI-stained bodies, focusing on differences in the frequency of univalents and bivalents. Each experiment was repeated three times and statistical analyses were conducted using R software, with a significance threshold set at $p < 0.05$.

### γ-irradiation
Worms of specified genotypes were exposed to 10 Gy of γ-irradiation using a $^{137}$Cs source (Gammacell1000 Elite; Nordion International). Analysis of diakinesis oocytes by DAPI staining was performed as described above at 28 hr post-irradiation.

### Immunostaining
Gonads from day 1 adults of the appropriate genotypes were dissected in 1X sperm salts (50 mM PIPES pH 7.0, 25 mM KCl, 1 mM MgSO$_4$, 45 mM NaCl, and 2 mM CaCl$_2$) with 1 mM levamisole and fixed in 1% paraformaldehyde diluted in 1X sperm salts for 5 min in a humid chamber. Slides were then frozen on a metal block on dry ice for at least 10 min prior to flicking off the cover slip and immersing in 100% ethanol at –20°C for 2 min. Slides were then washed 3X for 10 min each in PBSTB [1XPBS with 0.1% Tween and 0.1% bovine serum albumin (BSA)], and prior to overnight incubation with primary antibody also diluted in PBSTB. Primary antibodies were as follows: mouse anti-GFP 1:500 (Invitrogen Cat. #33-25—Lot 683990A), rabbit anti-phosphoHTP-3$^{S285}$ 1:2000 (*Das et al., 2022*), mouse anti-HA 1:1000 (mouse anti-HA-tag (6E2) Cell Signaling Cat. #23675, Lot 5); guinea pig anti-XND-1, 1:10,000 (*Wagner et al., 2010*). Overnight incubations were performed at 4°C except for anti-HA, which was performed at room temperature. The next day, slides were washed 3X in PBSTB for 10 min each and incubated with Alexa-conjugated secondary antibodies from Molecular Probes: anti-mouse Alexa 488, anti-rabbit Alexa 633, anti-guinea pig Alexa 568 (all diluted 1:2000 in PBST/BSA 0.1%).

Incubation was for 2 hr at room temperature in the dark. Slides were then washed 2 × 10 min in PBSTB, and 1 × 10 min with DAPI (10 mg/ml stock diluted 1:50,000 in 1X PBS). Slides were mounted in Prolong Diamond with DAPI and put in the dark to cure overnight before imaging. Images were acquired on a Leica Stellaris 5 confocal and Lightning image processing during acquisition. Z-stacks were acquired with 0.2 micron or smaller step size and visualized in 3D with Leica LAS X software. Each set of experiments was repeated three times.

All confocal microscopy images generated in this study have been deposited in the BioImage Archive under accession number S-BIAD2881 and S-BIAD2907.

## Y2H assay

cDNAs were amplified from germline-specific cDNA libraries and were cloned by Gibson Assembly into the *GAL4* activating domain (pGAD-C1) and binding domain (pGBD-C1)-expressing vectors. Plasmids with the desired cDNA were verified by Sanger sequencing prior to use. Plasmids were co-transformed into the *PJ69-4a* yeast strain and selected by growth on SC-LEU-TRP medium. Three to five transformants were grown to early log phase ($OD_{600}$ = 0.2), and 5 µl of culture was spotted onto selection plates:

- SC-LEU-TRP (loading control, selecting for plasmid presence)
- SC-LEU-TRP-HIS (reporter *HIS3* expression, indicating interaction)
- SC-LEU-TRP-ADE (more stringent selection for strong interactions)

Plates were incubated at 30°C for 72 hr and then imaged. The experiment was performed in triplicate, and empty vectors in the activating and binding domains were included as negative controls. Data are presented as spots on a plate and are from three to four individual transformants per interaction tested and are not individual colonies. The experiment was repeated in triplicate from different transformations.

## Co-immunoprecipitation and western blot

In order to perform co-immunoprecipitation experiments, synchronized *GFP::dsb-1; him-5::3xHA* and *him-5::3xHA* or *him-17::3xHA;Phim-5::him-5::GFP::3xFLAG* and *Phim-1::him-5::GFP::3xFLAG* strains were grown until young adult stage (24 hr post-L4) and nuclear protein fractionation was executed as previously shown (*Silva et al., 2014*). The chromatin-bound fraction was generated upon Benzonase digestion (25 U/100 µl of extract for 1 hr at 4°C). 2 mg of nuclear extract (nuclear-soluble and chromatin-bound fractions were pooled together) was incubated with Agarose GFP traps (Chromotek) or anti-HA Affinity Matrix (Roche) in Buffer D (20 mM HEPES pH 7.9, 150 mM KCl, 20% glycerol, 0.2 mM EDTA, 0.2% Triton X-100 and 1x complete Roche inhibitor) overnight at 4°C. The following day, the agarose beads were recovered by centrifugation at 7500 rpm for 2′ at 4°C and extensively washed with Buffer D at room temperature. After the final wash, the beads were resuspended in 40 µl of 2x Laemmli Buffer and boiled for 10′, after which they were spun at maximum speed for 1′ and the whole eluate was loaded on a precast 4–20% gradient acrylamide gel. Proteins were transferred on a nitrocellulose membrane for 90 min at 4°C at 100 V and blocked for 1 hr at room temperature in 1x TBST containing 5% BSA. Mouse monoclonal anti-HA (Cell Signaling), chicken polyclonal anti-GFP (AbCam), mouse monoclonal anti-FLAG HRP-conjugated (Sigma), and polyclonal anti-XND-1 (*Wagner et al., 2010*) antibodies were diluted in blocking solution at 1:1000, 1:5000, 1:2000, and 1:2500, respectively, and left to incubate overnight at 4°C. Washes were performed in 1x TBST and anti-mouse, anti-chicken, and anti-guinea pig HRP-conjugated secondary antibodies (Thermo Fisher) were diluted 1:10,000 in 1x TBST containing 5% milk for 1 hr at room temperature. After several washes in 1xTBST, the membrane was incubated with Clarity Max ECL (Bio-Rad) and imaged with a G-Box (Syngene).

## IP and MS analysis of HIM-17::GFP

IPs were performed on stock AV280: unc-119(e2498) III; him-17(ok424) V; meIs5[him-17::GFP +unc-119(+)]. Preparation and freezing of synchronized worms were carried out as in *Cheeseman et al., 2004* except that worms were grown on 8P plates seeded with the *E. coli* strain NA22 at room temperature. Worm lysis was carried out by grinding worm pellets in liquid nitrogen, and subsequent sonication (Diagenode) was performed with the following settings: 3x 5 cycles on high with 30 s on and 30 s off.

Lysate was centrifuged for 45 min at 15,000 rpm at 4°C. To avoid unspecific protein binding, the worm lysate was pre-cleared by incubation to uncoupled agarose beads (Chromotek, bab-20) for 1 hr at 4°C. Pre-cleared worm lysate was incubated with GFP-trap (Chromotek, gta-20) for 3 hr at 4°C. Beads were washed three times with lysis buffer containing proteinase inhibitors (Roche) and seven times with lysis buffer without proteinase inhibitors. Proteins were eluted by adding 40 µl 2x LSB and boiling for 5 min at 95°C. Protein precipitation was carried out using ProteoExtract Protein Precipitation Kit (EMD Millipore, 539180-1KIT) according to the manufacturer's protocol and submitted for MS analysis to the Taplin Biological Mass Spectrometry Facility. The MS proteomics data have been deposited to the ProteomeXchange Consortium via the PRIDE partner repository (*Perez-Riverol et al., 2022*) with the dataset identifier PXD057629 and 10.6019/PXD057629.

GFP pulldowns of HIM-17::GFP for IP/MS were performed in duplicate and corrected against unspecific binding of proteins bound to GFP trap beads. Specifically, potential false positive hits, defined as such because they were also detected in wild-type (N2), AIR-2::GFP and AIR-2KD::GFP pulldowns, were removed from this list. Then, only the remaining hits detected on both independent biological repeats, and for which 2 or more peptides were detected in each experiment, made it to this final table. The only exception is KU-80, which made it to the list after subtraction of the unspecific/false positive hits, but for which only 1 peptide was detected in the December pulldown (14 peptides were detected in the Feb 2014 pulldown). Given that we found KU-70 both times and the coverage/depth was better for the February 2014 pulldown, we decided to keep KU-80 on the final table. *Figure 1—figure supplement 2* shows all the hits that remained after subtraction of the unspecific binding proteins.

## Protein structure predictions

### Structure analysis of protein–protein interaction

Prediction of protein–protein interactions is based on available protein structures or AlphaFold2 (AF2) (*Jumper et al., 2021*) models. Most relevant proteins DSB-1, DSB-2, and DSB-3 in *C. elegans* do not have crystal structures. Hence, we rely on AF2 homology models that predict core domains with confidence factors better than 90, well within the range of applicability to scan for possible protein–protein interactions. Docked poses among these proteins will be predicted using ClusPro (*Comeau et al., 2004a*; *Comeau et al., 2004b*; *Kozakov et al., 2017*) and free energy complementarity will further analyze with FastContact (*Camacho and Zhang, 2005*; *Champ and Camacho, 2007*), both well-known structural modeling tools developed by one of us.

### Peptide design

Peptides were designed using MD simulations in AMBER18 on the GPU-accelerated code with the AMBER ff14SB force field (*Salomon-Ferrer et al., 2013*; *Maier et al., 2015*). The tLeap binary was used to solve structures in an octahedral TIP3P water box with a 15 Å distance from the peptide surface to the box edges and a closeness parameter of 0.75 Å. The system was neutralized and solvated in 150 mM NaCl. The nonbonded interaction cutoff was set to 8 Å. Hydrogen bonds were constrained using the SHAKE algorithm and an integration time step of 2 fs. Simulations were carried out by equilibrating the system for 5 ns at NPT, using a Berendsen thermostat to maintain a constant pressure of 1 atm followed by 300 ns NVT production at 300 K.

### Structural modeling of full-length DSB-1, DSB-2, and DSB-3 interactions

To model the interaction between the full-length DSB-1, DSB-2, and DSB-3 proteins, we used the Mol* Viewer (Molstar), an interactive web-based molecular visualization platform. Predicted structures were imported into the viewer to evaluate their relative orientation, potential interaction surfaces, and spatial compatibility. Full-length models were analyzed using chain-specific coloring and representation tools to facilitate comparative visualization and interpretation of possible interaction interfaces.

## Acknowledgements

We are grateful to P Carlton for the GFP::dsb-1 strain. This work was funded by R01GM104007, R01GM157825, and S10 grant 1S10OD030404 to JLY; the Czech Science Foundation (23-04918S) to NS; NIH grant R01GM072551 to MPC; R01ES030335 and Penn Center for Genome Integrity to KAB.

# Additional information

### Funding

| Funder | Grant reference number | Author |
|---|---|---|
| National Institutes of Health | R01GM072551 | Monica P Colaiacovo |
| National Institutes of Health | R01ES030335 | Kara A Bernstein |
| National Institutes of Health | R01GM104007 | Judith L Yanowitz |
| National Institutes of Health | S10OD030404 | Judith L Yanowitz |
| Czech Science Foundation | 23-04918S | Nicola Silva |
| National Institute of General Medical Sciences | R01GM157825 | Judith L Yanowitz |

The funders had no role in study design, data collection, and interpretation, or the decision to submit the work for publication.

### Author contributions

Marilina Raices, Conceptualization, Supervision, Investigation, Methodology, Writing – original draft, Writing – review and editing; Fabiola Balmir, Wei Li, McKenzie K Grundy, Dane K Hoffman, Elisabeth Altendorfer, Investigation; Nicola Silva, Funding acquisition, Validation, Investigation, Writing – original draft, Writing – review and editing; Carlos J Camacho, Software, Investigation, Methodology; Kara A Bernstein, Monica P Colaiacovo, Supervision, Funding acquisition, Investigation, Methodology, Writing – review and editing; Judith L Yanowitz, Conceptualization, Data curation, Supervision, Funding acquisition, Validation, Investigation, Methodology, Writing – original draft, Project administration, Writing – review and editing

### Author ORCIDs

Marilina Raices  https://orcid.org/0000-0001-8800-7116
Fabiola Balmir  https://orcid.org/0009-0003-8832-7197
Nicola Silva  https://orcid.org/0000-0001-5406-2280
McKenzie K Grundy  https://orcid.org/0000-0002-1995-1442
Elisabeth Altendorfer  https://orcid.org/0000-0002-9621-1632
Carlos J Camacho  https://orcid.org/0000-0003-1741-8529
Kara A Bernstein  https://orcid.org/0000-0003-2247-6459
Monica P Colaiacovo  https://orcid.org/0000-0001-7803-4372
Judith L Yanowitz  https://orcid.org/0000-0001-6886-8787

Joint Public Review: https://doi.org/10.7554/eLife.96458.4.sa1
Author response https://doi.org/10.7554/eLife.96458.4.sa2

# Additional files

### Supplementary files

Supplementary file 1. Strains and genetics. All strains were derived from the wild-type Bristol strain N2 and were cultivated at 20°C under standard conditions. Abbreviated names and full genotypes of the strains used in this study are listed here.

MDAR checklist

### Data availability

Mass spectrometry data was deposited in PRIDE. All data generated or analyzed during this study are included in the manuscript and supporting files; source data files have been provided for all figures.

**eLife** Research article

Developmental Biology

The following datasets were generated:

| Author(s) | Year | Dataset title | Dataset URL | Database and Identifier |
|---|---|---|---|---|
| Raices M, Balmir F, Silva N, Li W, Grundy MK, Hoffman DK, Altendorfer E, Camacho CJ, Bernstein KA, Colaiácovo MP, Yanowitz JL | 2025 | HIM-17-GFP LC-MS/MS analysis in *C. elegans* | https://www.ebi.ac.uk/pride/archive/projects/PXD057629 | PRIDE, PXD057629 |
| Raices M, Balmir F, Silva N, Li W, Grundy MK, Hoffman DK, Altendorfer E, Camacho CJ, Bernstein KA, Colaiácovo MP, Yanowitz JL | 2026 | HIM-17 and XND-1 localization in *C. elegans* | https://doi.org/10.6019/S-BIAD2881 | BioImage Archive, 10.6019/S-BIAD2881 |
| Raices M, Balmir F, Silva N, Li W, Grundy MK, Hoffman DK, Altendorfer E, Camacho CJ, Bernstein KA, Colaiácovo MP, Yanowitz J | 2026 | HIM-5 mislocalization in *dsb-1* mutants | https://doi.org/10.6019/S-BIAD2907 | BioImage Archive, 10.6019/S-BIAD2907 |

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
