## [Editor Report · eLife Assessment]

This study combines genetic, cell biological, and interaction data to propose a model of meiotic double-strand break regulation in *C. elegans*. **Solid** evidence supports the main conclusions, while by nature of a screening-type study, more may be needed to solidify speculations in future studies. Yet, comprehensive cataloging of the physical and genetic interactions of factors required for meiotic double-strand break is **useful** information for the field.

---

## [Referee Report · Joint Public Review]

Meiotic recombination begins with DNA double-strand breaks (DSBs) generated by the conserved enzyme Spo11, which relies on several accessory factors that vary widely across eukaryotes. In *C. elegans*, multiple proteins have been implicated in promoting DSB formation, but their functional relationships and how they collectively recruit the DSB machinery to chromosome axes have remained unclear.

In this study, Raices et al. investigate the biochemical and genetic interactions among known DSB-promoting factors in *C. elegans* meiosis. Using yeast two-hybrid assays and co-immunoprecipitation, they map pairwise protein interactions and identify a connection between the chromatin-associated protein HIM-17 and the transcription factor XND-1. They also confirm the established interaction between DSB-1 and SPO-11 and show that DSB-1 associates with the nematode-specific factor HIM-5, which is required for X-chromosome DSB formation.

The authors extend these findings with genetic analyses, placing these factors into four epistasis groups based on single- and double-mutant phenotypes. Together, these biochemical and genetic data support a model describing how these proteins engage chromatin loops and localize to chromosome axes. The work provides a clearer view of how *C. elegans* assembles its DSB-forming machinery and how this process compares to mechanisms in other organisms.

Comment from the Reviewing Editor on the revised version:

The authors have adequately addressed the prior review comments. At this point, after going through multiple rounds of reviews and revisions, the community will be better served by having this paper out in public. This version was assessed by the editors without further input from the reviewers.

---

## [Author Response]

The following is the authors’ response to the previous reviews

**Public Reviews:**

**Reviewer #1 (Public review):**
Summary:The manuscript by Raices et al., provides some novel insights into the role and interactions between SPO-11 accessory proteins in *C. elegans*. The authors propose a model of meiotic DSBs regulation, critical to our understanding of DSB formation and ultimately crossover regulation and accurate chromosome segregation. The work also emphasizes the commonalities and species-specific aspects of DSB regulation.Strengths:This study capitalizes on the strengths of the *C. elegans* system to uncover genetic interactions between a lSPO-11 accessory proteins. In combination with physical interactions, the authors synthesize their findings into a model, which will serve as the basis for future work, to determine mechanisms of DSB regulation.Weaknesses:The methodology, although standard, still lacks some rigor, especially with the IPs.
**Reviewer #2 (Public review):**
Summary:Meiotic recombination initiates with the formation of DNA double-strand break (DSB) formation, catalyzed by the conserved topoisomerase-like enzyme Spo11. Spo11 requires accessory factors that are poorly conserved across eukaryotes. Previous genetic studies have identified several proteins required for DSB formation in *C. elegans* to varying degrees; however, how these proteins interact with each other to recruit the DSB-forming machinery to chromosome axes remains unclear.In this study, Raices et al. characterized the biochemical and genetic interactions among proteins that are known to promote DSB formation during *C. elegans* meiosis. The authors examined pairwise interactions using yeast two-hybrid (Y2H) and co-immunoprecipitation and revealed an interaction between a chromatin-associated protein HIM-17 and a transcription factor XND-1. They further confirmed the previously known interaction between DSB-1 and SPO-11 and showed that DSB-1 also interacts with a nematodespecific HIM-5, which is essential for DSB formation on the X chromosome. They also assessed genetic interactions among these proteins, categorizing them into four epistasis groups by comparing phenotypes in double vs. single mutants. Combining these results, the authors proposed a model of how these proteins interact with chromatin loops and are recruited to chromosome axes, offering insights into the process in *C. elegans* compared to other organisms.Weaknesses:This work relies heavily on Y2H, which is notorious for having high rates of false positives and false negatives. Although the interactions between HIM-17 and XND-1 and between DSB-1 and HIM-5 were validated by co-IP, the significance of these interactions was not tested in vivo. Cataloging Y2H and genetic interactions does not yield much more insight. The model proposed in Figure 4 is also highly speculative.
**Reviewer #3 (Public review):**
The goal of this work is to understand the regulation of double-strand break formation during meiosis in *C. elegans*. The authors have analyzed physical and genetic interactions among a subset of factors that have been previously implicated in DSB formation or the number of timing of DSBs: CEP-1, DSB-1, DSB-2, DSB-3, HIM-5, HIM-17, MRE-11, REC-1, PARG-1, and XND-1.The 10 proteins that are analyzed here include a diverse set of factors with different functions, based on prior analyses in many published studies. The term "Spo11 accessory factors" has been used in the meiosis literature to describe proteins that directly promote Spo11 cleavage activity, rather than factors that are important for the expression of meiotic proteins or that influence the genome-wide distribution or timing of DSBs. Based on this definition, the known SPO-11 accessory factors in *C. elegans* include DSB-1, DSB2, DSB-3, and the MRN complex (at least MRE-11 and RAD-50). These are all homologs of proteins that have been studied biochemically and structurally in other organisms. DSB-1 & DSB-2 are homologs of Rec114, while DSB-3 is a homolog of Mei4. Biochemical and structural studies have shown that Rec114 and Mei4 directly modulate Spo11 activity by recruiting Spo11 to chromatin and promoting its dimerization, which is essential for cleavage. The other factors analyzed in this study affect the timing, distribution, or number of RAD-51 foci, but they likely do so indirectly. As elaborated below, XND-1 and HIM-17 are transcription factors that modulate the expression of other meiotic genes, and their role in DSB formation is parsimoniously explained by this regulatory activity. The roles of HIM-5 and REC-1 remain unclear; the reported localization of HIM-5 to autosomes is consistent with a role in transcription (the autosomes are transcriptionally active in the germline, while the X chromosome is largely silent), but its loss-of-function phenotypes are much more limited than those of HIM-17 and XND-1, so it may play a more direct role in DSB formation. The roles of CEP-1 (a Rad53 homolog) and PARG-1 are also ambiguous, but their homologs in other organisms contribute to DNA repair rather than DSB formation.

We appreciate the reviewer’s clarification. However, the definition of Spo11 accessory factors varies across the literature. Only Keeney and colleagues define these as proteins that physically associate with and activate Spo11 to catalyze DSB formation (Keeney, Lange & Mohibullah, 2014; Lam & Keeney, 2015). In contrast, other authors have used the term more broadly to refer to proteins that promote or regulate Spo11-dependent DSB formation, without necessarily implying a direct interaction with Spo11 (e.g., Panizza et al., 2011; Robert et al., 2016; Stanzione et al., 2016; Li et al., 2021; Lange et al., 2016). Thus, our usage of the term follows this broader functional definition.

An additional significant limitation of the study, as stated in my initial review, is that much of the analysis here relies on cytological visualization of RAD-51 foci as a proxy for DSBs. RAD-51 associates transiently with DSB sites as they undergo repair and is thus limited in its ability to reveal details about the timing or abundance of DSBs since its loading and removal involve additional steps that may be influenced by the factors being analyzed.

We agree with the reviewer that counting RAD-51 foci provides only an indirect measure of SPO-11–dependent DSBs, as RAD-51 marks sites of repair rather than the breaks themselves. However, we would like to clarify that our current study does not rely on RAD51 foci quantification for any of the analyses or conclusions presented. None of the figures or datasets in this manuscript are based on RAD-51 cytology. Instead, our conclusions are drawn from genetic interactions, biochemical assays, and protein–protein interaction analyses.

The paper focuses extensively on HIM-5, which was previously shown through genetic and cytological analysis to be important for breaks on the X chromosome. The revised manuscript still claims that "HIM-5 mediates interactions with the different accessory factors sub-groups, providing insights into how components on the DNA loops may interact with the chromosome axis." The weak interactions between HIM-5 and DSB-1/2 detected in the Y2H assay do not convincingly support such a role. The idea that HIM-5 directly promotes break formation is also inconsistent with genetic data showing that him5 mutants lack breaks on the X chromosomes, while HIM-5 has been shown to be is enriched on autosomes. Additionally, as noted in my comment to the authors, the localization data for HIM-5 shown in this paper are discordant with prior studies; this discrepancy should be addressed experimentally.

We appreciate the reviewer’s concerns regarding the interpretation of HIM-5 function. The weak Y2H interactions between HIM-5 and DSB-1 are not interpreted as direct biochemical evidence of a strong physical interaction, but rather as a potential point of regulatory connection between these pathways. Importantly, these Y2H data are further supported by co-immunoprecipitation experiments, genetic interactions, and the observed mislocalization of HIM-5 in the absence of DSB-1. Together, these complementary results strengthen our conclusion that HIM-5 functionally associates with DSB-promoting complexes.

Regarding HIM-5 localization, the pattern we observe using both anti-GFP staining of the eaIs4 transgene (Phim-5::him-5::GFP) and anti-HA staining of the HIM-5::HA strain is consistent with that reported by McClendon et al. (2016), who validated the same eaIs4 transgene. Although the pattern difers slightly from Meneely et al. (2012), that used a HIM5 antibody that is no longer functional and that has been discontinued by the commercial source. In this prior study, a weak signal was detected in the mitotic region and late pachytene, but stronger signal was seen in early to mid-pachytene. Our imaging— optimized for low background and stable signal—similarly shows robust HIM-5 localization in early and mid-pachytene, supporting the reliability of our GFP and HA-tagged analyses.

The recent analysis of DSB formation in *C. elegans* males (Engebrecht et al; PloS Genetics; PMID: 41124211) shows that in absence of him-5 there is a significant reduction of CO designation (measured as COSA-1 foci) on autosomes. This study strongly supports a direct and general role for HIM-5 in crossover formation— on both autosomes and on the hermaphrodite X.

This paper describes REC-1 and HIM-5 as paralogs, based on prior analysis in a paper that included some of the same authors (Chung et al., 2015; DOI 10.1101/gad.266056.115). In my initial review I mentioned that this earlier conclusion was likely incorrect and should not be propagated uncritically here. Since the authors have rebutted this comment rather than amending it, I feel it is important to explain my concerns about the conclusions of previous study. Chung et al. found a small region of potential homology between the *C. elegans* rec-1 and him-5 genes and also reported that him-5; rec-1 double mutants have more severe defects than either single mutant, indicative of a stronger reduction in DSBs. Based on these observations and an additional argument based on microsynteny, they concluded that these two genes arose through recent duplication and divergence. However, as they noted, genes resembling rec-1 are absent from all other Caenorhabditis species, even those most closely related to *C. elegans*. The hypothesis that two genes are paralogs that arose through duplication and divergence is thus based on their presence in a single species, in the absence of extensive homology or evidence for conserved molecular function. Further, the hypothesis that gene duplication and divergence has given rise to two paralogs that share no evident structural similarity or common interaction partners in the few million years since *C. elegans* diverged from its closest known relatives is implausible. In contrast, DSB-1 and DSB-2 are both homologs of Rec114 that clearly arose through duplication and divergence within the Caenorhabditis lineage, but much earlier than the proposed split between REC-1 and HIM-5. Two genes that can be unambiguously identified as dsb-1 and dsb-2 are present in genomes throughout the Elegans supergroup and absent in the Angaria supergroup, placing the duplication event at around 18-30 MYA, yet DSB-1 and DSB-2 share much greater similarity in their amino acid sequence, predicted structure, and function than HIM-5 and REC-1. Further, Raices place HIM-5 and REC-1 in different functional complexes (Figure 3B).

We respectfully disagree with the reviewer’s characterization of the relationship between HIM-5 and REC-1. Our use of the term “paralog” follows the conclusions of Chung et al. (2015), a peer-reviewed study that provided both sequence and microsynteny evidence supporting this relationship. While we acknowledge that the degree of sequence conservation is limited, the evolutionary scenario proposed by Chung et al. remains the only published framework addressing this question. Further the degree of homology between either HIM-5 or REC-1 and the ancestral locus are similar to that observed for DSB-1 and DSB-2 with REC-114 (Hinman et al., 2021). We therefore retain the use of the term “paralog” in reference to these genes. Importantly, our conclusions regarding their distinct molecular and functional roles are independent of this classification.

The authors acknowledge that HIM-17 is a transcription factor that regulates many meiotic genes. Like HIM-17, XND-1 is cytologically enriched along the autosomes in germline nuclei, suggestive of a role in transcription. The Reinke lab performed ChIP-seq in a strain expressing an XND-1::GFP fusion protein and showed that it binds to promoter regions, many of which overlap with the HIM-17-regulated promoters characterized by the Ahringer lab (doi: 10.1126/sciadv.abo4082). Work from the Yanowitz lab has shown that XND-1 influences the transcription of many other genes involved in meiosis (doi: 10.1534/g3.116.035725) and work from the Colaiacovo lab has shown that XND-1 regulates the expression of CRA-1 (doi: 10.1371/journal.pgen.1005029). Additionally, loss of HIM-17 or XND-1 causes pleiotropic phenotypes, consistent with a broad role in gene regulation. Collectively, these data indicate that XND-1 and HIM-17 are transcription factors that are important for the proper expression of many germline-expressed genes. Thus, as stated above, the roles of HIM-17 and XND-1 in DSB formation, as well as their effects on histone modification, are parsimoniously explained by their regulation of the expression of factors that contribute more directly to DSB formation and chromatin modification. I feel strongly that transcription factors should not be described as "SPO-11 accessory factors."

The ChIP analysis of XND-1 binding sites (using the XND-1::GFP transgene we provided to the Reinke lab) was performed, and Table S3 in the Ahringer paper suggests it is found at germline promoters, although the analysis is not actually provided. We completely agree that at least a subset of XND-1 functions is explained by its regulation of transcriptional targets (as we previously showed for HIM-5). However, like the MES proteins, a subset of which are also autosomal and impact X chromosome gene expression, XND-1 could also be directly regulating chromatin architecture which could have profound effects on DSB formation. As stated in our prior comments, precedent for the involvement of a chromatin factor in DSB formation is provided by yeast Spp1.

**Recommendations for the authors:**

**Editor comments:**
As you can see, the reviewers have additional comments, and the authors can include revisions to address those points prior to publicizing 'a version of record' (e.g. hatching rate assay mentioned by reviewer #1). This type of study, trying to catalog interactions of many factors, inevitably has loose ends, but in my opinion, it does not reduce the value of the study, as long as statements are not misleading. I suggest that the authors address issues by making changes to the main text. After the next round of adjustments by authors, I feel that it will be ready for a version of record, based on the spirit of the current eLife publication model.
**Reviewer #1 (Recommendations for the authors):**
I still have concerns about the HIM-17 IP and immunoblot probing with XND-1 antibodies. While the newly provided whole extract immunoblot clearly shows a XND-1 specific band that goes away in the mutant extracts, there is additional bands that are recognized - the pattern looks different than in the input in Figure 1B. Additionally, there is still a band of the corresponding size in the IPs from extracts not containing the tagged allele of HIM-17, calling into question whether XND-1 is specifically pulled down.The authors did not include the hatching rate as pointed out in the original reviews. In the rebuttal:"Great question. I guess we need to do this while back out for review. If anyone has suggestions of what to say here. Clearly we overlooked this point but do have the strain."

We thank the reviewer for this suggestion. We had intended to include a hatching analysis; however, during the course of this work we discovered that our him-17 stock had acquired an additional linked mutation(s) that altered its phenotype and led to inconsistent results. This strain was used to rederive the him-17; eaIs4 double mutant after our original did not survive freeze/thaw. Given the abnormal behavior observed in this line, we concluded that proceeding with the hatching assays could yield unreliable data. We are currently reestablishing a verified him-17 strain, but in the interest of accuracy and reproducibility, we have restricted our analysis in this manuscript to validated datasets derived from confirmed strains.

**Reviewer #2 (Recommendations for the authors):**
The authors have addressed most of the previous concerns and substantially improved the manuscript. The new data demonstrate that HIM-5 localization depends on DSB-1, and together with the Y2H and Co-PI results, strengthen the link between HIM-5 and the DSBforming machinery in *C. elegans*. The remaining points are outlined below:Specific comments:The font size of texts and labels in the Figure is very small and is hardly legible. Please enlarge them and make them clearly visible (Fig 1A, 1B, 2A, 2B, 2C, 2D, 2E, 3A, 3B, 3C, 3D, 3F)

Done

Although the authors have addressed the specificity of the XND-1 antibody, it remains unclear whether the boxed band is specific to the him-17::3xHA IP, since the same band appears in the control IP, albeit with lower intensity (Fig 1B). Is the ~100 kDa band in the him-17::3xHA IP a modified form XND-1? While antibody specificity was previously demonstrated by IF using xnd-1 mutants, it would be ideal to confirm this on a western blot as well.A Western Blot performed using whole cell extracts and probed with the anti- XND-1 antibody has been provided in the revised version of the manuscript (Fig. S1A). This confirms that the antibody specifically recognizes XND-1 protein. We believe that the ~100 kDa band mentioned by the reviewer is likely to be a non-specific cross reaction band detected by the antibody, since an identical band of the same mW was also detected in xnd-1 null mutants (Fig. S1A).Regarding the IP negative controls, we are firmly convinced the boxed band to be specific, and the fact that a (very) low intensity band is also found in the negative control should not infringe the validity of the HIM-17-XND-1 specific interaction. There is a constellation of similar examples present across the literature, as it is widely acknowledged amongst biochemists that some proteins may “stick” to the beads due their intrinsic biochemical properties despite usage of highly stringent IP buffers. However, the high level of enrichment detected in the IP (as also underlined by the reviewer) corroborates that XND-1 specifically immunoprecipitates with HIM-17 despite a low, non-specific binding to the HA beads is present. If interaction between XND-1 and HIM-17 was non-specific, we logically would have found the band in the IP and the band in the negative control to be of very similar intensity, which is clearly not the case.Although co-IP assays are generally considered not a strictly quantitative assay, we want to emphasize that a comparable amount of nuclear extract was employed in both samples as also evidenced by the inputs, in which it is also possible to see that if anything, slightly less nuclear extracts were employed in the him-17::3xHA; him-5::GFP::3xFLAG vs. the him5::GFP::3xFLAG negative control, corroborating the above mentioned points.Lastly, it is crucial to mention that mass spectrometry analyses performed on HIM17::3xHA pulldowns show XND-1 as a highly enriched interacting protein (Blazickova et al.; 2025 Nature Comms.), which strongly supports our co-IP results.The subheading "HIM-5 is the essential factor for meiotic breaks in the X chromosome" does not accurately represent the work described in the Results or in Figure 1. I disagree with the authors' response to the earlier criticism. The issue is not merely semantic. The data do not demonstrate that HIM-5 is required for DSB formation on the X chromosome - this conclusion can only be inferred. What Figure 1 shows is that XND-1 and HIM-17 interact, and that pie-1p-driven HIM-5 expression can partially rescue meiotic defects of him-17 mutants. This supports the conclusion that him-5 is a target of HIM-17/XND-1 in promoting CO formation on the X chromosome. However, the data provide no direct evidence for the claim stated in the subheading. I strongly encourage authors to revise the subheading to more accurately represent the findings presented in the paper.

After considering the reviewer’s comments, we have revised the subheading to more accurately describe our findings.

In Fig1C, please fix the typo in the last row - "pie1p::him5-::GFP" to "pie-1p::him- 5::GFP".

Done

In Fig 2C, "p" is missing from the label on the right for Phim-5::him-5::GFP.

Done

In Fig 3I, bring the labels (DSB-1/2/3) at the lower right to the front.

Done

In Concluding Remarks, please fix the typo "frequently".

Done

**Reviewer #3 (Recommendations for the authors):**
The experiments that analyze HIM-5 in dsb-1 mutants should be repeated using antibodies against the endogenous HIM-5 antibody, and localization of the HIM-5::HA and HIM-5::GFP proteins should be compared directly to antibody staining. This work uses an epitopetagged protein and a GFP-tagged protein to analyze the localization of HIM-5, while prior work (Meneely et al., 2012) used an antibody against the endogenous protein. In Figures 2 and S4 of this paper, neither HIM-5::HA nor HIM-5::GFP appears to localize strongly to chromatin, and autosomal enrichment of HIM-5, as previously reported for the endogenous protein based on antibody staining, is not evident. Moreover, HIM-5::GFP and HIM-5::HA look different from each other, and neither resembles the low-resolution images shown in Figure 6 in Meneely et al 2012, which showed nuclear staining throughout the germline, including in the mitotic zone, and also in somatic sheath cells. Given the differences in localization between the tagged transgenes and the endogenous protein, it is important to analyze the behavior of the endogenous, untagged protein. A minor issue: a wild-type control should also be shown for HIM-5::HA in Figure S4.

Wild type control added to figure S4

Evidence that XND-1 and HIM-17 form a complex is weak; it is supported by the Y2H and co-IP data but opposed by functional analysis or localization. The diversity of proteins found in the Co-IP of HIM-17::GFP (Table S2) indicate that these interactions are unlikely to be specific. The independent localization of these proteins to chromatin is clear evidence that they do not form an obligate complex; additionally, they have been found to regulate distinct (although overlapping) sets of genes. The predicted structure generated by Alphafold3 has very low confidence and should not be taken as evidence for an interaction.The newly added argument about the lack of apparently overlap between HIM-17 and XND1 due to the distance between the HA tag on HIM-17 and XND-1 is flawed and should be removed - the extended C-terminus in the predicted AlphaFold3 C-terminus of HIM-17 has been interpreted as if it were a structured domain. Moreover, the predicted distance of 180 Å (18 nm) is comparable to the distance between a fluorophore on a secondary antibody and the epitope recognized by the primary antibody (~20-25 nm) and is far below than the resolution limit of light microscopy.

We appreciate the reviewer’s thoughtful comment. The evidence supporting a physical interaction between XND-1 and HIM-17 is not only shown by our co-IP experiments, but it has also been recently shown in an independent study where MS analyses were conducted on HIM-17::3xHA pull downs to identify novel HIM-17 interactors (Blazickova et al.; 2025 Nature Comms). As shown in the data provided in this study, also under these experimental settings XND-1 was identified as a highly enriched putative HIM-17 interactor. We do acknowledge that their chromatin localization patterns are distinct and they regulate overlapping but not identical sets of genes, however, it is worth noting that protein–protein interactions in meiosis are often transient or context-dependent, and may not necessarily result in co-localization detectable by microscopy. In line with this, in the same work cited above, a similar situation for BRA-2 and HIM-17 was reported, as they were shown to interact biochemically despite the absence of overlapping staining patterns.

Minor issues:The images shown in Panel D in Figure 1 seem to have very different resolutions; the HTP3/HIM-17 colocalization image is particularly blurry/low-resolution and should be replaced. The contrast between blue and green cannot be seen clearly; colors with stronger contrast should be used, and grayscale images should also be shown for individual channels. High-resolution images should probably be included for all of the factors analyzed here to facilitate comparisons.